# Influence of Complexation with β- and γ-Cyclodextrin on Bioactivity of Whey and Colostrum Peptides

**DOI:** 10.3390/ijms241813987

**Published:** 2023-09-12

**Authors:** Tatsiana M. Halavach, Vladimir P. Kurchenko, Ekaterina I. Tarun, Natalia V. Dudchik, Mikalai M. Yatskou, Aleksey D. Lodygin, Ludmila R. Alieva, Ivan A. Evdokimov, Natasa Poklar Ulrih

**Affiliations:** 1Faculty of Biology, Belarusian State University, 220030 Minsk, Belarus; kurchenko@tut.by; 2Faculty of Environmental Medicine, International Sakharov Environmental Institute of Belarusian State University, 220070 Minsk, Belarus; ktarun@tut.by; 3Scientific Practical Centre of Hygiene, 220012 Minsk, Belarus; n_dudchik@mail.ru; 4Faculty of Radiophysics and Computer Technologies, Belarusian State University, 220030 Minsk, Belarus; yatskou@bsu.by; 5Faculty of Food Engineering and Biotechnologies, North Caucasus Federal University, Stavropol 355017, Russia; allodygin@yandex.ru (A.D.L.); lalieva@ncfu.ru (L.R.A.); ievdokimov@ncfu.ru (I.A.E.); 6Biotechnical Faculty, University of Ljubljana, 1000 Ljubljana, Slovenia; natasa.poklar@bf.uni-lj.si

**Keywords:** whey, colostrum, enzymatic hydrolysate, β-cyclodextrin, γ-cyclodextrin, inclusion complexes, antioxidant activity, antigenic properties, antimutagenic effect

## Abstract

Dairy protein hydrolysates possess a broad spectrum of bioactivity and hypoallergenic properties, as well as pronounced bitter taste. The bitterness is reduced by complexing the proteolysis products with cyclodextrins (CDs), and it is also important to study the bioactivity of the peptides in inclusion complexes. Hydrolysates of whey and colostrum proteins with extensive hydrolysis degree and their complexes with β/γ-CD were obtained in the present study, and comprehensive comparative analysis of the experimental samples was performed. The interaction of CD with peptides was confirmed via different methods. Bioactivity of the initial hydrolysates and their complexes were evaluated. Antioxidant activity (AOA) was determined by fluorescence reduction of fluorescein in the Fenton system. Antigenic properties were studied by competitive enzyme immunoassay. Antimutagenic effect was estimated in the Ames test. According to the experimental data, a 2.17/2.78-fold and 1.45/2.14-fold increase in the AOA was found in the β/γ-CD interaction with whey and colostrum hydrolysates, respectively. A 5.6/5.3-fold decrease in the antigenicity of whey peptides in complex with β/γ-CD was detected, while the antimutagenic effect in the host–guest systems was comparable to the initial hydrolysates. Thus, bioactive CD complexes with dairy peptides were obtained. Complexes are applicable as a component of specialized foods (sports, diet).

## 1. Introduction

Cyclodextrins (CDs) belong to a unique class of molecules that are naturally formed as a result of starch cleavage. CDs have a spatial cone-shaped structure with a hydrophobic cavity, which determines their ability to form stable inclusion complexes with various compounds in aqueous solutions [1,2,3]. Complexation is aimed at improving the physicochemical, and bioactive characteristics of food compounds (enhancement of solubility, stability and bioavailability). The use of inclusion complexes is aimed at increasing the shelf life of food products and masking undesirable organoleptic properties, as well as providing a controlled release of compounds [4,5].

α-, β-, and γ-CDs, formed by six, seven, and eight glucose units, are the most widespread and available in the group of CDs. Respectively, the number of glucose units determines the cavity size. Thus, α-CD is capable of forming inclusion complexes with low-molecular-weight substances or compounds with aliphatic side chains. β-CD interacts with aromatic compounds or heterocycles. γ-CD can accommodate a broader range of macromolecular organic compounds, including macrocycles and steroids [5]. Moreover, the water solubility of α-, β-, and γ-CDs is different. In particular, β-CD solubility in water at 25 °C is about 18.5 mg/mL, while that of α/γ-CD is 145/232 mg/mL, respectively [6]. It is known that γ-CD is actively hydrolyzed by salivary and pancreatic amylase, which are incapable of α- and β-CD cleaving. γ-CDs are rapidly digested and absorbed in the small intestine, unlike non-digestible α- and β-CD. In addition, β-CD (E_459_) is notable for its available price and active application in various industries [7].

α-, β-, and γ-CD do not have a toxic effect on the human organism when administered orally. There is a general trend at the legislative level toward wider recognition of these CDs as food additives [8,9,10]. The industrial production of cyclodextrins is well established, which determines their acceptable cost and availability for use in the food industry. In this regard, the prospect of using these compounds is associated with a high consumer demand for functional foods. In general, CDs can be used to obtain biotechnological compounds with innovative properties and to develop new food products [1,2].

Taste is an essential characteristic of food that determines its quality and consumer interest. Peptides are important in the formation of flavor characteristics of food, which defines the relevance of obtaining them from food sources as flavor enhancers [11]. It is known that food peptides have various bioactivities (immunomodulatory, antihypertensive, antioxidant, antimicrobial) along with a bitter taste, which prevents their use as functional nutrition ingredients and nutraceuticals. In this regard, encapsulation is a promising technology for the protection, sustained release, and delivery of bioactive peptides, enhancement of their physiological effects, and improvement of organoleptic and physicochemical properties [12].

Dairy protein hydrolysates, which are a mixture of peptides and amino acids, represent a valuable ingredient for dietary and sports nutrition products; however, due to a specific bitterness their use is limited. Bitterness is associated with the presence of specific amino acids (Phe, Trp, Pro, Ile, Tyr, His) and peptides with a molecular weight (M_W_) less than 6.5 kDa that contain these amino acids [13]. The masking of bitter taste is achieved by placing “bitter” amino acids and peptides in the CD cavity, which prevents binding to taste receptors. Trp, Tyr, and Phe interact with β-CD predominantly through the aromatic ring; hence, hydrophobic bonds are the main forces for the penetration of amino acids into the β-CD cavity. The participation of hydrogen bonds in maintaining the stability of amino acid complexes with β-CD was also confirmed [14]. In particular, the interaction of Trp, Tyr, Phe, and His with β-CD (encapsulation of the nonpolar part of the guest molecule) was revealed according to two-dimensional nuclear magnetic resonance (2D NMR) techniques (rotating frame Overhauser effect spectroscopy, ROESY), whereas the complexation was not demonstrated in the experiment with Pro and Ile [15].

There are known methods to reduce the bitterness of amino acids and enzymatic hydrolysates of soybean proteins by adding α-CD and β-CD [15,16,17,18], and also hydrolyzed proteins of rice [19], salmon [20], and pea [21] as a result of interaction with α-, β-, and γ-CD. At the same time, the effect of complexing conditions on the bitterness and hygroscopicity of γ-CD complexes with whey protein hydrolysate was studied [22]. According to our own experimental data, the specific features of whey and colostrum protein hydrolysis by endopeptidase alcalase were previously revealed and the bioactive properties of dairy hydrolysates were characterized [23,24,25]. Then, a reduction in bitterness, an increase in the antiradical properties, and the preservation of the antimutagenic activity of hydrolysed dairy proteins in inclusion complexes with β-CD were revealed [26,27,28]. It is important to compare the bioactive (antioxidant, antigenic, and antimutagenic) effect of whey and colostrum peptides contained in the host–guest systems with β- and γ-CD. The use of dairy peptides in bioactive inclusion complexes with cyclodextrins as a component of specialized foods is promising.

The aim of the current study is to optimize the conditions for obtaining inclusion complexes of β- and γ-CD with whey and colostrum peptides and to compare the biological activity of enzymatic hydrolysates and their β- and γ-CD complexes.

## 2. Results

### 2.1. Characteristics of Protein and Peptide Composition of Whey and Colostrum Hydrolysates

#### 2.1.1. High-Performance Liquid Chromatography (HPLC) Analysis of Hydrolysates and Their Ultrafiltrates

Experimental samples of enzymatic hydrolysates of dairy proteins (peptide fractions with a M_W_ less than or equal to 5 kDa) were analyzed by HPLC to determine their protein and peptide composition. Figure 1 and Figure 2 show HPLC profiles of the initial whey and colostrum protein hydrolysates and their filtrates.

The chromatographic profiles of dairy protein hydrolysates before and after filtration were compared. It was found that the fraction with a cutoff M_W_ of 5 kDa eluted from the column up to 23/25th min of separation of colostrum/whey hydrolysates. It is confirmed that fractionation leads to the removal of residual amounts of non-cleaved proteins and polypeptides with antigenic potential from enzymatic hydrolysates.

#### 2.1.2. Mass Spectrometric (MS) Study of Peptide Fractions

According to mass spectrometry results, an intermediate product of proteolysis with M_W_ more than 10 kDa was detected in the initial hydrolysates of whey and colostrum, as shown in Figure 3A and Figure 4A.

At the same time, peaks with mass/charge (*m*/*z*) values up to 5000 prevail in the spectra of filtrate of hydrolyzed whey (Figure 3B) and up to 4000 in the case of colostrum (Figure 4B). The data indicate the removal of the high-molecular-weight fraction after fractionation, as well as a greater depth of hydrolysis of colostrum proteins.

#### 2.1.3. Determination of Molecular Weight Distribution of Hydrolysates by Dynamic Light Scattering (DLS)

Analysis of whey and colostrum proteins hydrolysates was carried out using DLS method to determine particle sizes and their molecular weight distribution. A comparative study of filtrate samples with a 5 kDa molecular weight cutoff is presented in Table 1.

According to DLS data, fractions of cleaved whey and colostrum samples include a spectrum of peptides of similar size, which is evidenced by the relatively high polydispersity (Pd) of the system (Table 1). It should be noted that the colostrum hydrolysate contains a fraction with a smaller molecular weight (0.7–1.4 kDa) compared with the whey sample (1.4–2.3 kDa). In general, the average amino acid (AA) chain length in whey hydrolysate is 15 AA residues, whereas in the case of colostrum it is eight AA residues.

The data obtained are comparable with the mass spectrometry results (Figure 3 and Figure 4), which indicate a greater depth of hydrolysis of the colostrum protein component. This is due to the peculiarities of protein cleavage of casein and whey fractions by endopeptidase alcalase.

### 2.2. Characterization of Cyclodextrins Complexes with Dairy Peptides

#### 2.2.1. Thermal Degradation Parameters of Hydrolysates and Their Inclusion Complexes

Experimental samples of β/γ-CD complexes with whey and colostrum peptides were obtained at a CD:hydrolysate mass ratio of 2:1. An excessive amount of CD was used to achieve complete interaction between the peptides and the complexing agent. Thermogravimetric (TG) analysis was applied to demonstrate the formation of inclusion complexes. A comparative study of differential thermogravimetry (DTG) and differential scanning calorimetry (DSC) profiles obtained for control samples (peptides and β/γ-CD), mechanical mixtures made by blending the components at a mass ratio equal to 1:2, as well as experimental samples of complexes were performed.

Table 2 presents data on the thermal degradation parameters of pure substances (hydrolysate filtrates, β/γ-CD), mechanical mixtures of pure substances, and complexes according to DTG/DSC profiles.

The β/γ-CD sample is characterized by a mass loss peak at 301.8/303.4 °C with a maximum thermal destruction rate reaching 0.43/0.33 mg/°C (Figure 5 and Figure 6). In the case of whey protein hydrolysate, destruction peaks with maximum mass loss rate at 158.4, 206.3, 266.0, and 523.5 °C (0.005, 0.014, 0.031 and 0.054 mg/°C, respectively) were revealed. After thermal decomposition of colostrum hydrolysate, peaks with maximum mass loss rate at 159.0, 290.1, and 523.7 °C (0.015, 0.025, and 0.063 mg/°C, respectively) were detected.

There is a shifting of the β/γ-CD thermodestruction peak from 301.8/303.4 to 297.5/289.9 and 289.7/283.5 °C, respectively, in the analysis of mechanical mixtures containing whey and colostrum hydrolysate as reported in Table 2 and shown in Figure 5 and Figure 6. Along with this, the complex samples retained the dominant peak of β/γ-CD thermodegradation with a displacement and changing of its configuration, while the degradation peaks characteristic of the peptide mixture were practically not detected, supporting the complexes’ formation.

The case of the complexes prepared by mixing cyclodextrins and whey hydrolysate shows a shift in the β/γ-CD thermodestruction peak from 297.5/289.9 to 305.1/296.6 °C and a decrease in the sample mass loss speed from 0.29/0.17 to 0.15/0.11 mg/°C compared to the mechanical mixture, as shown in Figure 5. This demonstrates an enhanced resistance of the respective complex samples to thermal oxidative degradation. The decrease in the speed of thermodestruction from 0.30/0.18 to 0.14/0.15 mg/°C was established for the β/γ-CD complexes including colostrum hydrolysate, while T_Vmax_ (289.7 °C) was maintained for β-CD or while T_Vmax_ changed from 283.5 to 294.3 °C for γ-CD (Table 2, Figure 6B). Generally, an increase in the thermal oxidative stability of the peptide mixture in β/γ-CDs was proven.

Based on the activation energy (E_a_) estimates for mechanical mixtures and complexes, an increase in E_a_ by 1.2–1.5 and 1.6–1.9 times in relation to samples of whey and colostrum hydrolysates, respectively, was shown (Table 2). Thus, the stabilization of dairy peptide mixtures and β/γ-CD complexes was demonstrated.

As a result, the formation of inclusion complexes was proven by TGA data, and a higher stability of dairy peptides in interaction with cyclodextrins was demonstrated. The activation energy of β-CD was 324 kJ/mol, whereas that of γ-CD was 292 kJ/mol, which confirms the greater thermal stability (increase in the activation energy of thermal oxidative degradation) of the β-form. Moreover, the E_a_ of β-CD complexes with whey and colostrum peptides reached 105 and 107 kJ/mol, respectively, which exceeds the values obtained for complexes with γ-CD (97 and 102 kJ/mol, respectively).

In whole, the main advantage of experimental samples with γ-CD is its better solubility. The technological process of complexes based on obtaining γ-CD does not require heating to increase its solubility, which is relevant for β-CD. The advantages of β-CD are its relative thermostability, a more pronounced sweet taste, a simple chemical synthesis scheme, and an available price category. β-CD is registered as food additive E_459_ and is actively used in different industries.

#### 2.2.2. Fluorescence Emission Spectrometry of β/γ-CD Complexes

The results of determining the optimal conditions for β/γ-CD interaction with dairy peptides are presented. The fluorescence emission spectra of Trp residues in whey protein hydrolysate revealed an increase in the signal level with higher β/γ-CD content.

The dependences of Trp fluorescence level in whey protein hydrolysate on the mass ratio of β/γ CD:hydrolysate were obtained as shown in Figure 7. In particular, an increase in Trp fluorescence intensity was found when the content of β- and γ-CD in the whey protein hydrolysate solution was raised to a WH–5 kDa:β-CD mass ratio and WH–5 kDa:γ-CD mass ratio of 1:0.75 and 1:1, respectively. Further addition of β/γ-CD to the hydrolyzed whey did not reveal a significant change in the spectral indices. It should be noted that the fluorescence intensity of samples with β-CD was higher, whereas a smaller effect was observed in experiments with γ-CD.

Similar studies were performed for a mixture of colostrum protein hydrolysate with β/γ-CD. Some increase in Trp fluorescence intensity was found with a higher β-CD content. Due to the relatively low level of Trp fluorescence in the composition of colostrum peptides, it seemed difficult to perform quantitative calculation of the experimental data.

According to the experimental data, a mass ratio of WH–5 kDa:β/γ-CD equal to 1:1 (a small molar excess of the complexing agent is assumed) was defined for the subsequent determination of the optimal time for formation of whey peptide complexes with cyclo-dextrins. A more pronounced increase in fluorescence intensity was observed when β-CD was added to the whey peptide mixture compared to γ-CD (Figure 8A).

Incubation of hydrolyzed whey proteins with β- and γ-CDs for 15 min revealed an increase in fluorescence to the maximum level (Figure 8B). When the samples were incubated for up to 240 min, no significant change in fluorescence was detected, suggesting that an equilibrium system state was established after 15 min of component interaction.

The duration of the technological process is recommended to be 30 min for the most complete interaction of β- and γ-CD with whey peptides. In general, according to fluorescence spectroscopy data, the optimum parameters for the interaction of whey protein hydrolysate with β/γ-CD to obtain complexes were established. These are WH–5 kDa:β/γ-CD mass ratio of 1:1 and an incubation time of 30 min. The above parameters are also recommended for obtaining β/γ-CD complexes with colostrum hydrolysate.

#### 2.2.3. Determination of Molecular Weight Distribution of Cyclodextrin Complexes with Whey Peptides According to DLS

Complexes of β/γ-CD with whey peptides were analyzed using the DLS method to determine particle sizes and their molecular weight distribution. A comparative study of whey protein hydrolysate and corresponding inclusion complexes depending on the incubation time of peptides with β/γ-CD was presented (Table 3).

The initial cyclodextrin is present in each sample independent of the time of measurement. The signal from β/γ-CD particles almost completely eliminates the signal from hydrolyzed protein particles when CD is double in excess by mass.

Both the initial β-CD and peptides and/or their complexes are present in the solution at the mass ratio WH–5 kDa:β-CD equal to 1:0.5 and 1:1. The formation of β-CD and WH–5 kDa complexes with the particle size of 1.0–1.4 nm, which corresponds to the estimated molecular weight of 3.1–7.2 kDa, is assumed. The complex detection was observed at 15 min after mixing. The interaction of hydrolysate and β-CD with the maximum amount of cyclodextrin (1:2) revealed complex formation only in the samples obtained at 15 min after mixing.

Both the initial γ-CD and the peptides and/or their complexes were detected in the solution in the case of the WH–5 kDa:γ-CD mass ratio of 1:0.5 and 1:1. Complexation of peptides and γ-CD with a particle size of 1.0–1.1 nm (3.2–4.6 kDa) is implied after 15 min of interaction.

In general, the DLS method allows to confirm the CDs complex formation with peptides of whey proteins at the mass ratio of components equal to 1:0.5 and 1:1. At the same time, the signal from cyclodextrin complexes practically completely neutralizes the signal from hydrolysate at two-fold excess of CD in the mixture.

Thus, according to the comparison results of DLS method and fluorescence spectroscopy data, the mass ratio of WH–5 kDa:β/γ-CD equal to 1:1 and incubation of the mixture for 30 min is optimal for achieving the most complete interaction of cyclodextrins with peptides. The above parameters have also been suggested for obtaining complexes with colostrum hydrolysate.

### 2.3. Effect of Complex Formation on Antioxidant Properties of Cleaved Dairy Proteins

The antioxidant activity (AOA) of native whey and colostrum proteins, their enzymatic hydrolysates and ultrafiltrates, and β- and γ-CD complexes with peptides was defined. Dependences of fluorescein (FL) fluorescence intensity on the concentration of native dairy proteins and their hydrolysates, complexes, and cyclodextrins were obtained. Studies were performed in a wide range of concentrations of the studied compounds (0.3–400 µg of protein/mL). Sample concentrations corresponding to 50% recovery of FL fluorescence on addition of experimental samples (IC_50_ values) were determined.

A significant increase in the radical reducing properties of whey/colostrum hydrolysates was demonstrated after alcalase proteolysis and ultrafiltration. This effect is shown in Table 4. The calculation of IC_50_ values for the content of protein component revealed the increase in antioxidant potential of hydrolyzed and fractionated whey and colostrum samples by 2.54–5.91 and 3.29–6.62 times, respectively. The radical reducing activity of whey protein peptides was 1.6 times lower than that of similar colostrum samples. According to mass spectrometry and DLS data, a higher degree of hydrolysis is characteristic of colostrum samples, which is associated with effective cleavage of casein into low-molecular-weight peptides, while whey proteins are relatively stable to proteolysis.

The present study also revealed the effect of complexation with β/γ-CD on the antiradical activity of whey and colostrum peptides (WH–5 kDa and CH–5 kDa). According to the fluorimetric method, the AOA of colostrum/whey hydrolysate in the complexes with β-CD increased by 1.45/2.18 times, in the experiment with γ-CD—by 2.14/2.78 times.

The increase in the antioxidant properties of the obtained complexes might be related to the higher solubility of the peptide fraction in the host–guest systems with β- and γ-CD. In addition, it should be noted that the antioxidant potential of γ-CDs is greater compared to β-CD (Table 4), apparently due to the relatively high solubility of γ-CD. Interaction with γ-CD has a greater effect on increasing the AOA of whey and colostrum peptides than that of β-CD. At the same time, complexation with cyclodextrins is more effective in increasing the AOA of whey peptides rather than of colostrum.

In the whole, a different level of AOA of native and cleaved whey and colostrum proteins is caused by the degree of hydrolysis as well as by the features of protein and peptide composition (mass ratio of casein and whey protein fractions) and the content of nonprotein antioxidants. The fluorimetric method revealed a significant increase in the AOA of colostrum and whey hydrolysates as part of complexes with β/γ-CD.

### 2.4. Antigenic Properties of Peptide Fractions and Their Complexes with Cyclodextrins

The results of the antigenic properties evaluation (ability to bind with antibodies) of whey/colostrum proteins and their enzymatic hydrolysates as well as β/γ-CD complexes with peptide fractions are reported. The residual amount of the main milk allergen (β-lactoglobulin, β-lg) in hydrolyzed and fractionated samples and in experimental complex samples was determined by competitive enzyme-linked immunosorbent assay (ELISA).

The residual antigenicity (relative content of native β-lg) decreased at 17.2 and 147.1-fold, respectively, after proteolysis of whey and colostrum proteins with alcalase, as shown in Table 5. Along with this, the antigenicity of the hydrolyzed whey protein sample subjected to filtration with a molecular weight cutoff equal to 5 kDa decreased by 2 × 10^3^ times. No β-lg was detected in the fractionated hydrolyzed colostrum sample.

Thus, as a result of fractionation, the residual antigenicity of whey and colostrum hydrolysates is maximally reduced due to the removal of proteolysis products with a molecular weight greater than 5 kDa that contain antigenic determinants. The greater antigenicity of whey protein hydrolysates is due to the high initial content of β-lg (50% of whey proteins).

The present study also established the effect of complexation with β/γ-CD on the antigenic properties of whey and colostrum peptides. In the experiment with control samples, the addition of β/γ-CD did not affect the binding of β-lg to antibodies. For β/γ-CD complexes with whey protein hydrolysate, a five-fold decrease in the antigenic properties of the peptides was found.

In general, the different level of residual antigenicity of native and cleaved samples of whey and colostrum proteins is determined by the features of their protein and peptide composition (the mass ratio of casein and whey fractions). According to competitive ELISA data, the inhibition of antibody binding to whey peptides in the composition of complexes with β/γ-CD was established, probably related to the shielding of binding sites with antibodies against β-lg.

### 2.5. Complexation Effect on Antimutagenic Action of Dairy Peptides

The antimutagenic effect of peptide fractions of hydrolyzed whey and colostrum and their complexes with cyclodextrins was evaluated. The antimutagenic potential of the experimental samples was evaluated in the Ames test based on consideration of the frequency of reverse mutations to histidine prototrophy for *S. typhimurium* strains.

It was found that the hydrolysate and complex samples in the concentration range of 0.03–0.5 mg protein per plate did not exhibit a bacteriostatic/bactericidal effect against *S. typhimurium* TA 98 and TA 100, which potentially caused false-positive results.

The amount of revertants in the negative control was within the range of fluctuations of the spontaneous level for these strains. The response of strains to standard mutagens (positive control) was found to be within standard levels. Differences between the amount of revertants in the experiment were statistically negligible in relation to the negative control at *p* < 0.05.

Statistically significant decrease in induced mutation was observed for tested hydrolysates and complexes. The revealed differences in the amount of revertants in the control and experimental samples were statistically significant in the range of concentrations studied (0.03–0.5 mg of protein per plate). Along with this, control samples of β/γ-CD possessed no antimutagenic activity.

According to the data presented in Table 6, the decrease in the level of induced mutation when experimental samples were added at the maximum concentration (0.5 mg/protein per plate) reached 21.28–29.86% in the test system with *S. typhimurium* TA98 and 19.36–22.97% in the experiment with *S. typhimurium* TA100. It was found that the difference in the level of antimutagenic activity of peptides of whey and colostrum is not significant.

The antimutagenic effect of the colostrum samples did not significantly differ when tested against *S. typhimurium* TA98 and TA100 strains, which is reflected in Table 6. In the case of whey hydrolysate and the corresponding γ-CD complexes, a greater level of mutation reduction was established for the test system with *S. typhimurium* TA98. It should be particularly noted that the experimental samples of hydrolysates and β/γ-CD complexes in the range of studied concentrations have comparable antimutagenic effects.

In general, as a result of complexation with cyclodextrins, the antimutagenic effect of whey and colostrum peptides remains at the initial level.

## 3. Discussion

According to current data, enzymatic hydrolysis is the most common method for improving the bioactive characteristics of dairy proteins, which depend on the specifics of the proteases used (substrate and site specificity, reaction conditions) [29,30]. In the technological treatment and digestion process in the gastrointestinal tract, dairy proteins are cleaved into peptides with a broad spectrum of bioactive functions (antimicrobial, antioxidant, opioid immunomodulatory activity) [31]. In addition, potential antibody binding sites are cleaved as a result of hydrolysis, thereby preventing allergic reactions [32]. Thus, the technological process of hydrolysate production includes the choice of substrate and enzyme (enzymes), conditions of enzymatic reaction, and characterization of proteolysis products [29,30,32].

A well-known proteolytic enzyme is alcalase, an extracellular bacterial protease isolated from the culture medium of *Bacillus subtilis*. The enzyme is actively used in the modification of food products, in particular, to reduce their allergenic potential [33,34]. Peptides with antioxidant, hypotensive, metal-binding, antidiabetic, anti-inflammatory, and antimicrobial activities were found among the products of alcalase hydrolysis. The broad substrate and site specificity of alcalase suggests the use of this enzyme for hydrolysis of various protein substrates with a high degree of proteolysis and predominance of low-molecular-weight peptides [35].

Based on the results of experimental study, enzymatic hydrolysates of dairy proteins (whey and colostrum) were obtained using bacterial endopeptidase alcalase and subsequent ultrafiltration with a molecular weight cutoff of 5 kDa. The peptide profile of hydrolysate samples was analyzed using HPLC, mass spectrometry, and DLS method. A greater depth of hydrolysis of colostrum proteins was established, which is caused by effective proteolysis of casein in colostrum and relative resistance of whey proteins to alcalase cleavage.

Hydrolysates contain a wide range of peptides differing in amino acid chain length, hydrophobicity, and charge. Separation methods are aimed at obtaining peptide fractions with a target range of physicochemical and functional properties. In particular, a mixture of peptides in the process of membrane ultrafiltration is separated according to their molecular weight [35,36]. It should be noted that the classification of hypoallergenic infant formulae is based on the molecular weight distribution of the peptides. Partially hydrolyzed mixtures (allergy prevention) mainly include peptides with an M_W_ of up to 10 kDa, whereas mixtures based on extensive hydrolysates (clinical nutrition) are represented by a fraction with an M_W_ of less than 3 kDa [37].

In the present study, both highly active endopeptidase alcalase and subsequent ultrafiltration were used to achieve effective hydrolysis of whey and colostrum proteins as well as to remove the residual amount of undigested protein substrate. Subsequently, inclusion complexes of dairy peptides with natural cyclodextrins (β/γ-CD) were made.

According to literature data, the effect of CDs complexation with protein hydrolysates of animal and plant origin on changes in organoleptic properties of included peptides was studied [15,16,17,18,19,20,21,22]. Thus, α-CD was first applied to change the taste of amino acids (Arg, Phe, Val, Leu, Ile) and synthetic peptides. It has been reported that an excessive amount of α-CD has to be introduced to perceptibly reduce the bitter taste of the compounds tested [16]. Inclusion of α-CD (10%) in soy hydrolysate solution (5% of protein) was enough to considerably reduce the bitter taste (from 10 to four points) [17]. Along with this, the addition of β-CD (5%) to the soy hydrolysate solution (5% of protein) was responsible for the maximum bitterness reduction (up to 1 point) [15]. These are the first reports on the application of α/β-CD to reduce the bitterness of soy protein hydrolysate containing a mixture of peptides [15,16,17]. Subsequently, the interaction of peptides in the enzymatic hydrolysis of soybean meal with α-, β-, and γ-CDs was estimated. It was found that β-CD (mass ratio of protein:CD equal to 1:2) is the best alternative for adding to protein hydrolysate from soybean meal due to a pronounced decrease in bitterness and lower production cost [18].

It is important to note the study on the interaction between α/β/γ-CD (3 and 5%) and rice protein peptides (5%). Based on the organoleptic analysis data, the bitterness of the samples increased in the sequence: 5% β-CD < 3% β-CD < 5% γ-CD < 3% γ-CD < 5% α-CD < 3% α-CD < hydrolysate. The addition of β/γ-CD to rice protein hydrolysate caused a significant reduction in its bitterness, with β-CD appearing stable in the gastrointestinal tract [19].

Subsequently, the effect of 2-butanol and β-CD on the bitterness, and the physical, chemical, and functional properties of salmon protein hydrolysate obtained using alcalase was revealed. The maximum effect on bitterness reduction was achieved after hydrolysate treatment with β-CD at a mass ratio of 1:1 for 30 min [20]. In addition, the solutions of pea hydrolysate (100 g/L) when treated with 2-butanol and β-CD (15 g/L) had a minimum level of bitterness [21].

The formation of CD complexes with amino acids was confirmed by nuclear magnetic resonance (NMR) spectroscopy [14,16,17,19,21], differential scanning calorimetry [14], fluorescence spectroscopy [19], and Fourier-transform infrared spectroscopy [20,21]. NMR spectra of the soy hydrolysate sample indicated a strong interaction between the aromatic and aliphatic regions of the peptides and the internal cavities of natural CDs [18].

The influence of the complexation conditions on the properties of γ-CD complexes with whey protein hydrolysate was investigated [22]. The effect of temperature (20–50 °C), incubation time (3–9 h), and the mass ratio of γ-CD to peptides (1:4, 2:3, and 3:2) on complex formation was established. Hydrolysate with minimal bitterness was obtained at 35 °C for 6 h and γ-CD:hydrolysate mass ratio of 3:2. This variant is the most similar to the method of obtaining complexes proposed in the present work. At the same time, several differences related to the conditions of complexes producing of γ-CD with dairy peptides are discovered.

Thus, the presented work determined the optimal hydrolysate:CD mass ratio of 1:1 and sufficient incubation time (30 min) for the effective interaction of peptides with β/γ-CD. Complexation was proved by TGA/DSC analysis, fluorescence spectroscopy, and DLS method.

According to previous studies, a comparative analysis of hydrolyzed and fermented proteins of whey and colostrum was performed [25], and an increase in the antiradical effect [23] and antimutagenic action [24,25] with a rise in the degree of hydrolysis of protein substrates was revealed. Further, a reduction in bitterness, an increase in antioxidant activity, and the persistence of antimutagenic activity of dairy peptides in β-CD complexes were detected [26,27,28]. The main objective of this paper was to compare the bioactivities (antioxidant, antigenic, and antimutagenic effects) of whey and colostrum peptides before and after interaction with β- and γ-CD. AOA study of the whey and colostrum peptide fractions and their complexes with β- and γ-CD was performed.

It is known that the antiradical effect of milk is defined by the properties of the protein fraction (casein, whey proteins) [38]. The bioactivity of dairy protein hydrolysates is determined by the presence of certain amino acids in the peptide sequence. Thus, the sulfhydryl group of Cys directly interacts with radicals. The imidazole group of His is associated with chelation of metal ions, hydrogen transfer, or radical capture, whereas aromatic amino acids (Tyr and Try) have proton donor properties [38,39].

Several recent works contain data on the AOA increase in dairy proteins subjected to hydrolysis by various proteases (alcalase, trypsin, pepsin, neutrase, flavourzyme, etc.) [40,41]. Alcalase-derived peptides have relatively high antioxidant properties [39,42].

The effect is known of complexation with cyclodextrins on the AOA level of a wide range of natural antioxidants using various methodological approaches, in particular 2,2′-azino-bis(3-ethylbenzothiazoline-6-sulfonic acid (ABTS)–Trolox assay, oxygen radical absorbance capacity (ORAC) assay, 1,1-diphenyl-2-picrylhydrazyl (DPPH) radical scavenging assay, ferric reducing antioxidant power (FRAP) [43,44,45,46,47,48].

It was found that stable inclusion complexes of β-CD with the key coffee compounds (caffeic and quinic acids) which impart bitterness, astringency, and acidity to coffee are useful for improving bioactivity and masking the bitter taste of various products [43]. Complexation of β-CD with the alkaloid piperine provides an increase in its bioavailability and AOA [44]. According to the results of in vitro experiments (DPPH method), the AOA of the bioflavonoid chrysin enhanced in the host–system with β-CD due to its solubility increase, while the cytotoxicity of complexes was significantly elevated towards three human cancer cell lines (A549, HT–29 and HCT116) [45]. According to the results of in vitro studies (DPPH, ABTS and FRAP), the complexation of catechin (as a component of green tea) with β-CD provides an increase in radical reduction activity, along with an improvement in complex stability [46]. The complexes containing p-anisaldehyde (a biodegradable agent for food preservation) showed better stability during storage in low-humidity conditions, higher AOA, and a comparable level of antimicrobial activity against *Escherichia coli* and *Staphylococcus aureus* [47]. It was noted that food nanocarriers containing antimicrobial and antioxidant agents are promising candidates for increasing the storage period of food products and substituting synthetic preservatives [48].

The present study demonstrated an increase in the antioxidant potential of whey and colostrum protein hydrolysates in complexes with β/γ-CD, which probably results from the increased solubility of the peptide fraction. It was reported that interaction with γ-CD had a greater effect on the AOA increase of whey and colostrum peptides compared to β-CD. This effect is assumed to be caused by the relatively high solubility of γ-CD.

At the next stage, the antigenic properties of experimental samples of hydrolysates and corresponding CD complexes were studied. The relevance of this study is associated with the introduction of hydrolysates into specialized hypoallergenic infant formulas [49,50], sports, and clinical nutrition [51,52]. After cleavage of dairy proteins by flavourzyme, alcalase, and protozyme, an increase in the degree of hydrolysis and cleavage of the main allergen proteins (casein, β-lg and α-la), along with the appearance of a tart and bitter taste, was found [53]. The potential use of immobilized alcalase to produce hypoallergenic hydrolysates was demonstrated, but free alcalase was more effective in cleaving α-la and β-lg [54].

β-lg is one of the main dairy allergens [55], and sensitization to it is caused by numerous epitopes located along the entire length of the molecule [56,57]. According to Commission Directive 2006/141/EC, the immunoreactive β-lg content in infant food should be less than 1%. In general, the known IgE and IgG binding epitopes of β-lg [57] contain cleavage sites by alcalase, potentially resulting in an effective reduction of its antigenicity. Moreover, CDs are immunologically safe for use as delivery systems for various bioactive substances in animals [58,59].

The antigenicity of whey protein hydrolysate subjected to filtration (5 kDa) decreased by 2 × 10^3^ times, which corresponds to the requirements for extensive hydrolysates for therapeutic foods in the experiment [37]. β-lg was not detected in the colostrum peptide fraction. The inhibition of antibody binding to whey peptides in complexes with β/γ-CD was established. This effect is probably due to the masking of binding sites with antibodies against β-lg.

Particular interest is expressed in the antimutagenic potential of fermented dairy products [60], as well as hydrolysates of food proteins [61]. According to literature data, the antimutagenic activity of proteinogenic amino acids, except for Cys, was studied against the mutagen N-methyl-N′-nitro-N-nitrosoguanidine (MNNG) using *Salmonella typhimurium* TA100 indicator strain in the Ames test [62]. Maximum activity was shown for Cys (IC_50_—0.003 μmol per plate), while Gly, Trp, Lys, and Arg also had relatively high antimutagenic effects (IC_50_—0.08–0.13 μmol per plate). It is assumed that the antimutagenicity is indirectly caused by the interaction of the α-amino group of the amino acids with MNNG through the formation of N-nitroso compounds. The degree of inhibition also differs significantly depending on the structure of the amino acid radicals. Thus, the activity of Cys was more than 200-fold higher than that of Ala without the SH-group. It appears that the SH-group of Cys undergoes rapid methylation due to the capture of methyl radicals formed from MNNG [62]. The antimutagenic effect of casein was studied against a range of mutagens (benzopyrene, N-methylnitrosourea, 4-nitrobenzoyl chloride, N-nitroquinoline-1-oxide). The antimutagenicity rate of casein increased significantly after pepsinolysis, which is associated with the release of reactive peptides [63].

Experimental hydrolysates of whey and colostrum proteins had a significant antimutagenic action in the Ames test, while no relevant change in tested bioactivity of peptide fractions in the host–guest systems was found. At the same time, an increase in antiradical activity as well as a reduced antigenic potential of dairy peptides during complexation were revealed.

It should be particularly noted that in the presented work the alternative to whey proteins is the application of primary milk (colostrum). Colostrum provides a rich natural source of macro- and micronutrients, immunoglobulins, peptides with antimicrobial activity, and growth factors [64,65]. At present, food products and nutritional additives based on primary milk are used as a general health-enhancing agent and to support the immune status of the human body [66]. Experimental enzymatic hydrolysates of colostrum were proven to possess biologically active (antioxidant, antimutagenic) effects comparable to those of whey proteins.

It is intended to release data on the comparative study of β/γ-CD complexes on the organoleptic properties of enzymatic hydrolysates of whey proteins and colostrum with an extensive degree of hydrolysis. Experiments are planned to study the antibacterial activity of β/γ-CD complexes against Gram-positive and Gram-negative pathogens.

The experimental data are a contribution to the improvement of the technology of producing hypoallergenic hydrolysates of dairy proteins with proven biologically active properties. The novelty of this work consists in the determination of the complexation effect with β/γ-CD on the antioxidant, antigenic, and antimutagenic properties of whey and colostrum peptides.

## 4. Materials and Methods

### 4.1. Preparation of Whey and Colostrum Protein Hydrolysates and Determination of Their Protein and Peptide Composition

#### 4.1.1. Enzymatic Hydrolysis and Fractionation of Proteolysis Products

The concentrate of bovine whey proteins (technical specifications BY 100377914.550–2008), skim bovine colostrum powder (All-Russian Research Institute of Dairy Industry, Moscow, Russia), enzyme Alcalase^®^ 2.4 L FG (EC 3.4.21.62, protease from *Bacillus licheniformis*, 2.4 U/g, Novozymes, Copenhagen, Denmark), and phosphate buffered saline (PBS, pH 7.4) from Sigma (St. Louis, MO, USA) were used in this study.

Solutions of whey and colostrum (5% per protein content) in 50 mM PBS were prepared. The resulting solutions were centrifuged at 10,000 g for 30 min to remove insoluble particles, and the supernatant was used for hydrolysis. Enzymatic cleavage was performed at an enzyme/substrate mass ratio of 5%, a temperature of 50 °C, and an active acidity of the medium of 7.4 pH units for 2 h during hydrolysis of whey proteins and 3 h in the experiment with colostrum. Spin-X UF Concentrator 20 filters (Corning, London, UK) with a cutting capacity of 5 kDa were used for hydrolysate fractionation. Mass fraction of total protein in filtrate samples was determined according to ISO 8968–1:2014 [67]. Peptide fractions were subsequently used to obtain complexes with cyclodextrins.

The hydrolysate samples were frozen at −20 °C and part of the samples were freeze-dried for the following thermogravimetric analysis.

#### 4.1.2. High-Performance Liquid Chromatography of Hydrolysates

Acetonitrile (ACN) for HPLC (Sigma Aldrich, St. Louis, MO, USA) and trifluoroacetic acid (TFA) for mass spectrometry (Fisher Scientific International, Hampton, NH, USA) were used in the present studies. Standards of dairy proteins (bovine origin) were β-lg (variants A and B, 90% of protein), α-la (85% of protein), BSA (90% of protein), casein (88% of protein), and Ig G (90% of protein) from Sigma (St. Louis, MO, USA).

HPLC analysis was performed on Agilent 1100 chromatograph (Agilent, Santa Clara, CA, USA) using Zorbax–300SB C8 column (4.6 × 250 mm, 5 µm, Agilent, Santa Clara, CA, USA). The column was equilibrated with 0.1% aqueous TFA solution. The samples were eluted using acetonitrile gradient (ACN-water-TFA = 95:5:0.1 mL/100 mL): 0–5 min, 5%; 5–10 min, 5–10%; 10–30 min, 10–40%; 30–32 min, 40%; 32–40 min, 40–50%; 40–45 min, 50%; 45–50 min, 50–10%. Separation was performed at room temperature with a flow of 1.0 mL/min for 50 min, and detection was performed at 214 nm. HPLC profiles were analyzed using specialized software ChemStation for LC 3D systems Rev.B.04.01 (Agilent, Santa Clara, CA, USA).

#### 4.1.3. Mass Spectrometric Analysis of Hydrolysates

MS analysis was performed using purified water (Honeywell Burdick & Jackson, Muskegon, MI, USA) and TFA for mass spectrometry (Fisher Scientific International, Hampton, NH, USA), Protein Calibration Standard I/Standard II, and α-cyano-4-hydroxycinnamic acid ionization matrix (Bruker Daltonik, Bremen, Germany).

Thawed samples (according to Section 4.1.1.) were centrifuged at 10,000× *g* for 2 min, and TFA solution (2%) was added to the supernatant at a volume ratio of 19:1. The obtained solutions were centrifuged at 10,000× *g* for 2 min. The supernatants were desalted and concentrated using Supel Tips C18 pipette tips (Supelco, Bellefonte, PA, USA) containing C18 reverse-phase sorbent based on recommendations of the product manufacturer. The matrix and sample eluates were applied to the MALDI plate at a volume ratio of 1:1 and air-dried.

A high-resolution mass spectrometer with MALDI (matrix-activated laser desorption ionization) ionization source in combination with a time-of-flight (TOF) mass analyzer—microflex LRF MALDI–TOF (Bruker Daltonics, Bremen, Germany) was used in this research. The resolution was over 15,000 FWHM at *m*/*z* 500–5000 Da in the reflectron mode, the error reached 500 ppm in the linear mode at *m*/*z* 5–100 kDa, and 30 ppm in the reflectron mode at *m*/*z* 500–5000 Da. Compass for flexSeries 1.4 and flexControl Version 3.4 (Build 57), Bruker Daltonics BioTools 3.2 SR4 and Build 6.32 (Bruker Daltonik, Bremen, Germany) software were used for data processing.

### 4.2. Preparation of β- and γ-Cyclodextrin Complexes with Whey and Colostrum Protein Hydrolysates and Their Characteristics

#### 4.2.1. Incubation of Cyclodextrins with Whey and Colostrum Peptides

Filtrates of whey and colostrum hydrolysates prepared according to Section 4.1.1, β-CD (Roquette, Lestrem, France), and γ-CD (Chem-Impex International Inc., Jinan, China) were used to obtain CD complexes with peptides.

Solutions containing β/γ-CD and hydrolysates (peptide fractions) in 50 mM PBS at different mass ratios of CD:protein (2:1, 1:1) were incubated at 25 and 50 °C for β- and γ-CD, respectively, and stirred continuously for 30–60 min. The components were mixed at a 2:1 mass ratio of CD:protein to obtain samples for the thermogravimetric analysis, whereas the 1:1 mass ratio was used in experiments on studying bioactivities (AOA, antigenic and antimutagenic effects). The process involving β-CD was carried out at an increased temperature (50 °C) in order to maximize the solubility of the cyclodextrin. Subsequently, the samples were stored at −20 °C, and part of the samples were freeze-dried for further thermogravimetric analysis.

#### 4.2.2. Fluorescence Emission Spectrometry (Fluorescence Emission Spectra of Tryptophan Residues)

Whey and colostrum protein hydrolysates (peptide fractions) obtained according to Section 4.1.1 were used in the experiment. Solutions containing hydrolysates (5 mg protein/mL) and varying amounts of β/γ-CD (0–10.0 mg/mL) in deionized water-based 50 mM PBS were prepared. Then, 200 μL of samples including hydrolysate:CD at a mass ratio of 1:0.25/0.5/0.75/1.0/2.0 were added to 96-well black plate (Corning, London, UK) and incubated at 25 °C for 0–240 min with permanent shaking.

The fluorescence intensity of tryptophan residues was measured using a Varian Cary Eclipse spectrofluorimeter (Varian Inc., Palo Alto, CA, USA) at the fluorescence excitation wavelength (λ_ex_) of 295 nm and emission intensity (λ_em_) in the range of 305–450 nm (slit width 5 nm). Specialized software Cary Eclipse WinUV 4.2 (Agilent, Santa Clara, CA, USA) was used to process the results of scanning.

#### 4.2.3. Analysis of Hydrolysates/Complexes by Dynamic Light Scattering

This experiment used whey/colostrum protein hydrolysates (filtrates with a molecular weight cutoff of 5 kDa) obtained according to Section 4.1.1., β-CD (Roquette, Lestrem, France), and γ-CD (Chem-Impex International Inc., Jinan, China), deionized water-based 50 mM PBS.

The DLS method is founded on measuring the fluctuations of intensity of scattered light, passed through the studied solution, that allows to calculate diffusion coefficients of particles in the solution. According to the obtained values of diffusion coefficients, the size of particles and their size distribution are found and then the molecular weight of these particles is estimated. The method is designed for solutions of macromolecules with a molecular weight greater than 1 kDa.

Samples containing hydrolysates and CDs in 50 mM PBS were prepared as described:(1)WH–5 kDa (1.0 mg/mL);(2)CH–5 kDa (1.0 mg/mL);(3)β-CD (0.5/1.0/2.0 mg/mL);(4)γ-CD (0.5/1.0/2.0 mg/mL);(5)β-CD:WH–5 kDa = 0.5:1.0 (0.5 mg/mL of β-CD and 1.0 mg/mL WH–5 kDa);(6)β-CD:WH–5 kDa = 1.0:1.0 (1.0 mg/mL of β-CD and 1.0 mg/mL WH–5 kDa);(7)β-CD:WH–5 kDa = 2.0:1.0 (2.0 mg/mL of β-CD and 1.0 mg/mL WH–5 kDa);(8)γ-CD:WH–5 kDa = 0.5:1.0 (0.5 mg/mL of γ-CD and 1.0 mg/mL WH–5 kDa);(9)γ-CD:WH–5 kDa = 1.0:1.0 (1.0 mg/mL of γ-CD and 1.0 mg/mL WH–5 kDa);(10)γ-CD:WH–5 kDa = 2.0:1.0 (2.0 mg/mL of γ-CD and 1.0 mg/mL WH–5 kDa).

The measurements were performed on a DynaPro^®^ NanoStar^®^ instrument (Wyatt Technology, Goleta, CA, USA). Fifty µL of the sample was injected into a plastic disposable cuvette. Measurement was carried out under following conditions: cuvette compartment temperature equaled to 25 °C, number of measurements was 10, measurement time 20 s. The molecular weight of the particles in the sample solution was estimated in DYNAMICS 7.8.1 program (Wyatt Technology, Goleta, CA, USA). The “Coils” model was used to calculate data for the analysis of hydrolyzed protein particles, and the “Rayleigh Spheres” model was used for cyclodextrins and complexes.

#### 4.2.4. Thermogravimetric Analysis and Differential Scanning Calorimetry of Inclusion Complexes

β-CD (Roquette, Lestrem, France), γ-CD (Chem-Impex International Inc., Jinan, China), freeze-dried samples of hydrolysates (Section 4.1.1), and CD complexes with peptides (Section 4.2.1) were used for TGA/DSC analysis. The study was performed on a TGA/DSC I thermogravimetric analyzer (Mettler Toledo, Greifensee, Switzerland).

Weighted control samples (hydrolysates and β/γ-CD), mechanical mixtures of hydrolysates and β/γ-CD at a mass ratio of 1:2, and experimental samples of complexes were prepared. The pure substances were 20 mg weighted. The mechanical mixtures contained 6.7 mg of whey/colostrum hydrolysate and 13.3 mg of β/γ-CD (mass ratio of 1:2).

A 20 mg sample was used in the TGA/DSC experiments, which were performed in the temperature range from 30 to 600 °C, the heating rate was 5 °C/min, the resolution equaled 1 µg, and the accuracy of temperature control was ±2 °C.

Data were obtained in the form of mass loss curve (thermogravimetry, TGA) and a curve of the sample mass change rates as a function of system temperature (differential thermogravimetry, DTG).

Thermal decomposition stages were fixed for each sample under programmed heating conditions from 30 °C to 600 °C at a rate of 5 °C/min. The activation energy (E_a_) was measured according to the Broido methods using the TGA curves [68]. A comparative study of the TGA/DSC profiles of control samples (peptide fractions and β/γ-CD), mechanical mixtures prepared by blending the components at a mass ratio of 1:2, and test samples of complexes were performed.

### 4.3. Evaluation of Antioxidant Activity of Enzymatic Hydrolysates and Cyclodextrin Complexes with Peptides

The decrease in fluorescence intensity of fluorescein (FL), due to its binding with oxygen radicals, was used to estimate the antioxidant activity of experimental samples. The approach used in this work is based on AOA determining of the samples according to their ability to bind free radicals formed in the Fenton system.

Fluorescein, Mora salt, ethylenediaminetetraacetic acid (EDTA), and phosphate buffered saline (PBS, pH 7.4) obtained from Sigma (St. Louis, MO, USA), and hydrogen peroxide (Acros Organics, Geel, Belgium) were used in the experiment.

In addition, 2 µM FL solution, 250 mM Mora salt solution, 50 mM EDTA solution, 1.0 mM Mora salt (Fe^2+^) solution with 1.0 mM EDTA, 10 mM H_2_O_2_ solution, 50 mM PBS (pH 7.4), β/γ-CD solutions (3–4000 μg/mL), the test samples of hydrolysates (Section 4.1.1), and inclusion complexes (Section 4.2.1) with protein concentration equal to 3–4000 μg/mL in 50 mM PBS were prepared.

A 0.02 μM FL solution in 50 mM PBS was added to a cuvette with an optical path length of 1 cm. Fluorescence measurements were performed on RF–5301 PC fluorimeter (Shimadzu, Kyoto, Japan) at an excitation wavelength (λ_ex_) of 490 nm and emission intensity (λ_em_) in the range of 495–600 nm (slit width 5 nm). The maximum fluorescence intensity was recorded at λ_514_; the fluorescence intensity of the obtained peak was taken as the maximum (100%).

A 0.02 μM FL solution in 50 mM PBS containing 0.1 mM Fe^2+^ with 0.1 mM EDTA and 1.0 mM H_2_O_2_ was added to the cuvette. When Fe^2+^ ions interacted with H_2_O_2_ (Fenton reaction), the resulting radicals suppressed the FL fluorescence. The obtained value of the fluorescence intensity of the obtained peak was taken as the minimum.

A 0.02 μM FL solution in 50 mM PBS containing 0.1 mM Fe^2+^ with 0.1 mM EDTA and 0.3–400 μg/mL (per protein/cyclodextrin amount) of experimental samples were added to the cuvette. The reaction was started by adding H_2_O_2_ at a final concentration of 1.0 mM. The obtained values of fluorescence peaks were expressed as a percentage, taking as 100% the fluorescence of the control FL solution. The relative fluorescence intensity (A, %) was calculated from Equation (1):(1)$$A=\frac{Fl}{Fl_{0}}\times 100,$$
where *Fl*_0_ is the fluorescence intensity of the control FL sample (FL solution without Fe^2+^, EDTA, hydrolysate, and H_2_O_2_); Fl is the fluorescence intensity of the solution after antioxidant adding. 

Graphs of fluorescence intensity dependence (A, %) on protein content in the analyzed samples were plotted. According to the equation obtained, the concentration of sample IC_50_ corresponding to 50% fluorescence quenching was calculated.

### 4.4. Antigenicity Determination of Dairy Peptides and their Complexes with Cyclodextrins

The kit for quantitative determination of β-lg by competitive enzyme immunoassay (RIDASCREEN^®^β-Lactoglobulin, R4901) from R-Biopharm AG (Darmstadt, Germany) was used to evaluate the antigenic properties of experimental samples. It is designed to analyze the β-lg content in hydrolyzed dairy products, including hypoallergenic infant formula. The kit was calibrated with control samples of hydrolyzed β-lg and the standards consisted of native allergen protein. The test system allows the detection of native and treated protein, as well as its fragments. The detection limit is 2.1 µg/g of β-lg and the limit of quantification is 5.0 µg/g of β-lg.

Dilutions of native and hydrolyzed whey and colostrum proteins (Section 4.1.1), and their complexes with CDs (Section 4.2.1) and β/γ-CD (control) were prepared according to the test system manufacturer’s recommendations. Cyclodextrin content in complexes was equal to the protein concentration of the whey/colostrum samples. The optical density of the target solutions was determined using Multiskan Ascent microplate spectrophotometer (Thermo Labsystems, Waltham, MA, USA). Experimental data were processed with RIDA^®^SOFT Win.net 1.103.0.0217 program (R-Biopharm AG, Darmstadt, Germany) using cubic spline function.

Residual antigenicity (RA) was calculated as the ratio of β-lg content in the hydrolysed sample to that in the initial (non-treated) sample and expressed in %.

### 4.5. Evaluation of Antimutagenic Effect of Native Hydrolysates and in Complexes with Cyclodextrins

Indicator strains of *Salmonella typhimurium* TA98 and TA100 from the collection of Scientific Practical Centre of Hygiene (Minsk, Belarus) were taken as test models in a short-term test to study antimutagenic properties. Ethidium bromide for *S. typhimurium* TA98 and sodium azide for *S. typhimurium* TA100 produced by Sigma (St. Louis, MO, USA) were used as direct mutagens. The antimutagenic activity of the hydrolysates and their complexes was evaluated in a modified Ames test (the methodology is presented in [24]).

The samples of β/γ-CD (control), hydrolysates, and CD complexes obtained according to Section 4.1.1 and Section 4.2.1, respectively, were filtered through syringe filters (0.45 μm, Carl Roth, Karlsruhe, Germany) to remove microbiological contamination. The antimutagenic effect of the studied hydrolysate and complex samples was evaluated when 10 μg of mutagens per plate were added to the test system. The protein/cyclodextrin concentration was 0.03–0.50 mg per plate during the test. The experiment was followed by positive controls. Three replicates were used in each control and experimental variants.

The level of mutation reduction (*I_m_*, %) was calculated by using the Equation (2):(2)$$I_{m}=100-\frac{N_{1}}{N_{2}}\times 100,$$
where *N*_1_ is the number of revertants in the experiment, *N*_2_ is the number of revertants in the positive control.

### 4.6. Statistical Analysis

A one-, two-, or three-way analysis of variance (ANOVA) [69] followed by Dunnett’s test [70], Student’s *t*-test, or Tukey’s Honest Significant Difference (HSD) test [71,72] were used to compare the means of factor’s levels. Dunnett’s test was applied for comparing several treatments with a control at the family-wise confidence level 0.95. The two-sample Student’s *t*-test was used to check whether the unknown means of two groups are equal or not. The Tukey’s HSD test computed the differences between the means of the levels of a factor with the specified family-wise confidence level (0.95). All the statistical analyses were conducted using the R functions *aov*, *DunnettTest*, *t.test*, and *TukeyHSD* of the packages *stats* [73] and *DescTools* [74]. Statistical differences between group means were considered significant at the level *p* < 0.05, adjusted for the multiple pairwise comparisons.

Graphs were plotted in Microsoft Office 2021 Excel (MS Corporation, Shadeland, IN, USA). Experimental results in tables and graphs are presented as the mean ± the half-width of 95% confidence interval (n = 3).

## 5. Conclusions

Inclusion complexes of β- and γ-cyclodextrin with enzymatic hydrolysates (peptide fractions) of whey and colostrum proteins were prepared under optimized conditions in the present study. The effect of cyclodextrin complexation on the antioxidant properties, antigenic potential, and antimutagenic effect of the included dairy peptides was determined. In particular, an increase in the antioxidant effect of hydrolysates in the host–guest systems with β- and γ-cyclodextrin was demonstrated using the fluorimetric method. The competitive ELISA test revealed a decrease in the antigenic potential of whey peptides as a result of complexation. The Ames test showed a comparable antimutagenic effect for the original peptides and the corresponding complexes. Thus, the obtained inclusion complexes of β- and γ-cyclodextrin with whey and colostrum peptides have a confirmed bioactive action. They are a promising ingredient for functional foods.

## Figures and Tables

**Figure 1 ijms-24-13987-f001:**
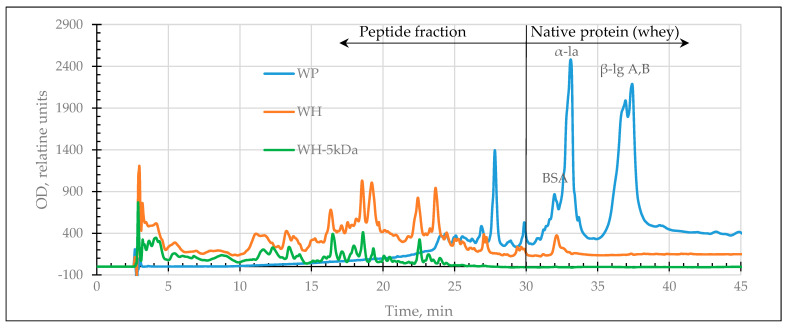
HPLC profiles of native (WP), hydrolyzed whey proteins (WH), and their filtrate (WH–5 kDa). α-la—α-lactalbumin, β-lg A,B—β-lactoglobulin (variants A,B), BSA—bovine serum albumin.

**Figure 2 ijms-24-13987-f002:**
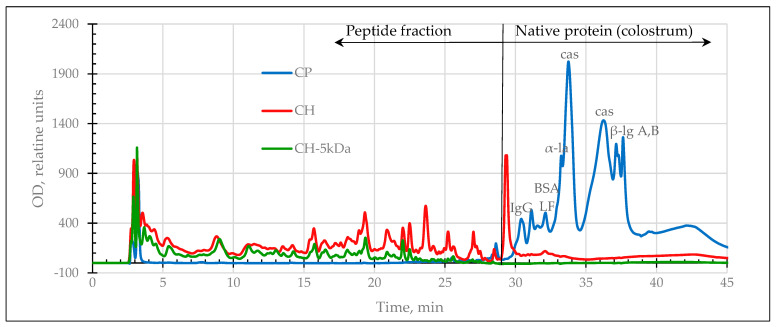
HPLC profiles of native (CP), hydrolyzed colostrum proteins (CH), and their filtrate (CH–5 kDa). α-la—α-lactalbumin, β-lg A,B—β-lactoglobulin (variants A,B), BSA—bovine serum albumin, LF—lactoferrin, cas—casein, IgG—immunoglobulin G.

**Figure 3 ijms-24-13987-f003:**
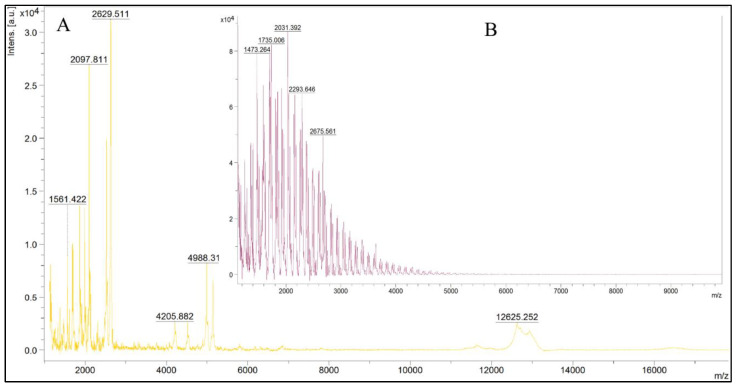
MS profiles of hydrolyzed whey proteins (**A**) and their ultrafiltrate (**B**).

**Figure 4 ijms-24-13987-f004:**
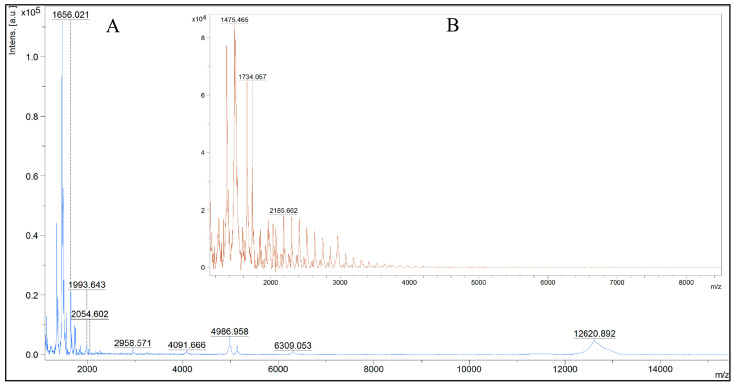
MS profiles of hydrolyzed colostrum proteins (**A**) and their ultrafiltrate (**B**).

**Figure 5 ijms-24-13987-f005:**
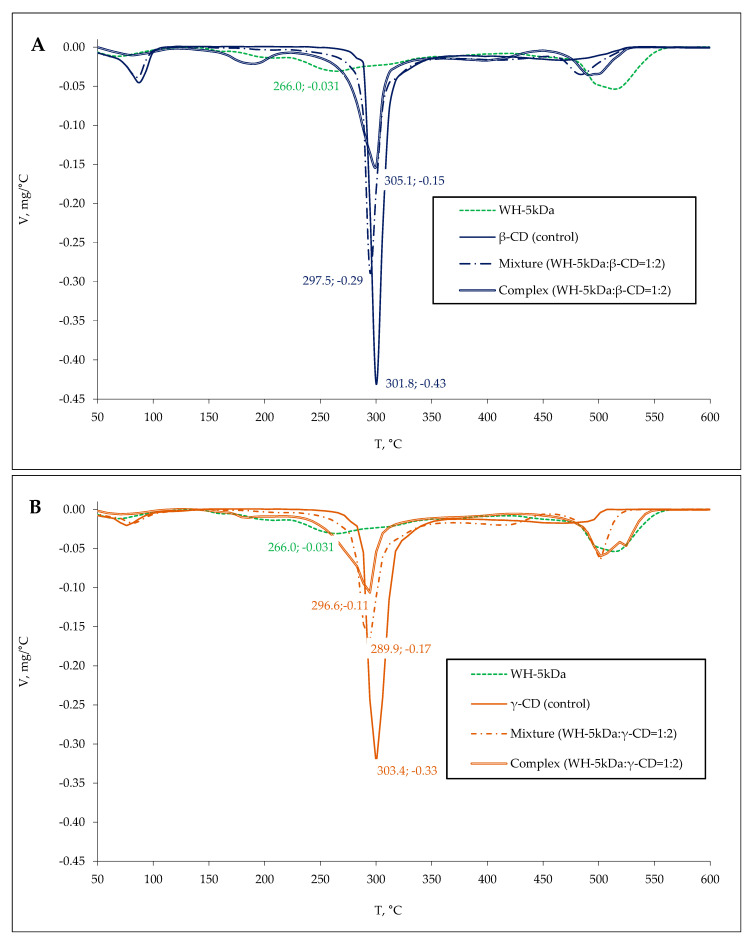
DTG spectra of whey-based control and experimental samples with β-CD (**A**) and γ-CD (**B**). WH–5 kDa—ultrafiltrate of whey hydrolysate.

**Figure 6 ijms-24-13987-f006:**
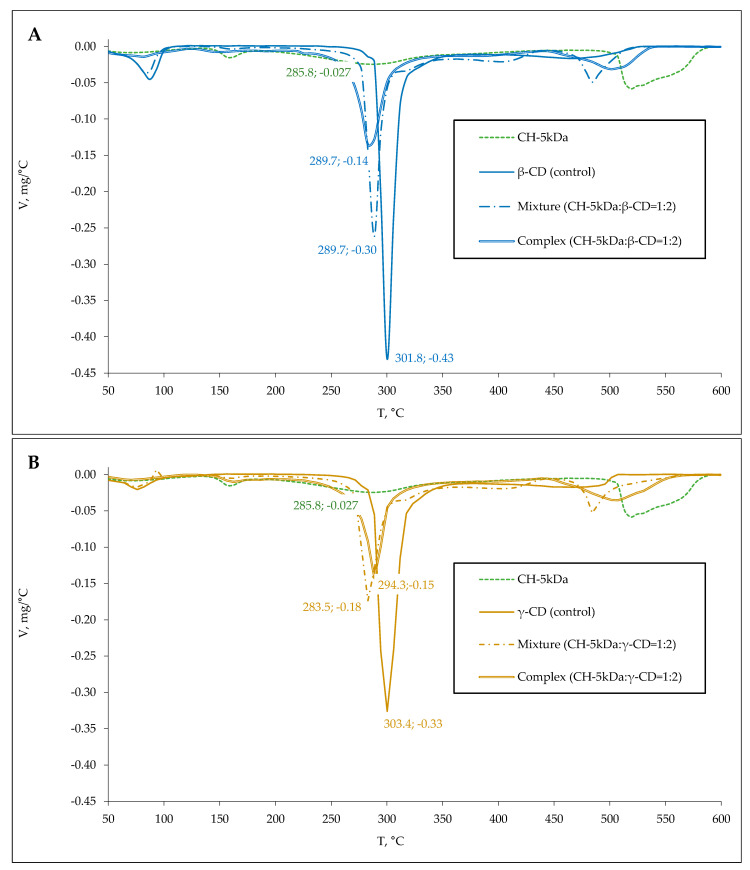
DTG spectra of colostrum-based control and experimental samples with β-CD (**A**) and γ-CD (**B**). CH–5 kDa—filtrate of colostrum hydrolysate.

**Figure 7 ijms-24-13987-f007:**
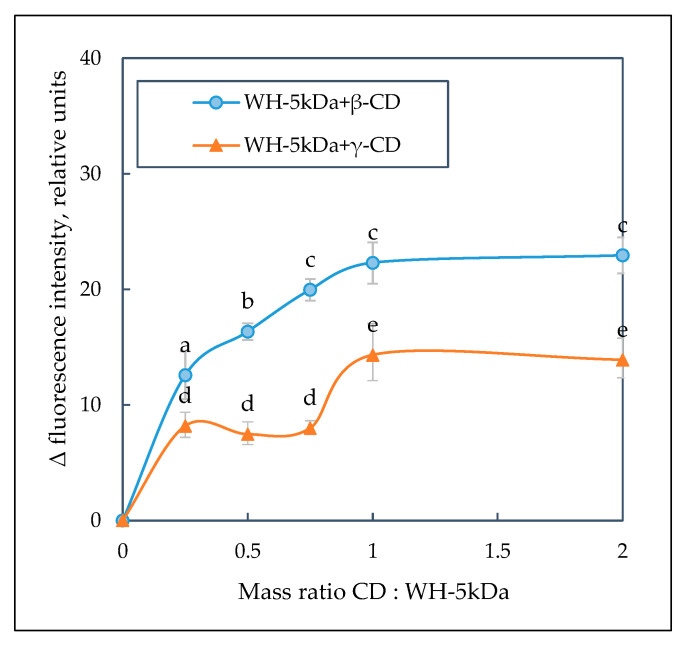
The dependence of Trp fluorescence intensity on WH–5 kDa:β/γ-CD mass ratio. The values represent the mean ± the half-width of 95% confidence interval (n = 3). Means without a common letter (a–e) on the graph points indicate significant difference at *p* < 0.05.

**Figure 8 ijms-24-13987-f008:**
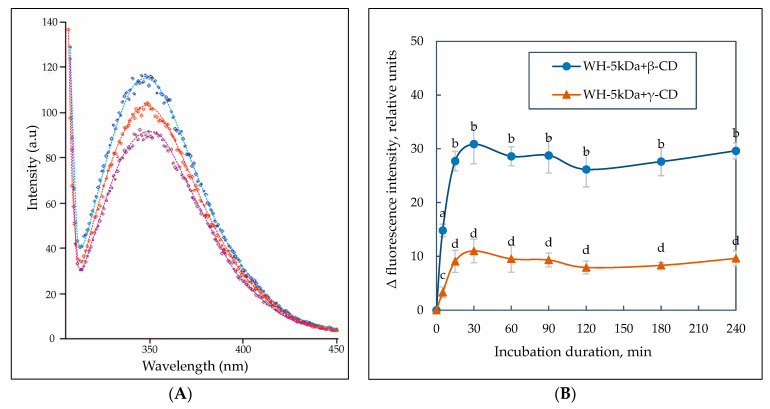
Fluorescence emission spectra of Trp residues at WH–5 kDa:CD mass ratio of 1:1 (from bottom to top—control, complex with γ- and β-CD) after 15 min incubation of hydrolysate with CD (**A**). Trp fluorescence intensity dependence on the incubation time of WH–5 kDa with β/γ-CD (**B**). The values represent the mean ± the half-width of 95% confidence interval (n = 3). Means without a common letter (a–d) on the graph points indicate significant difference at *p* < 0.05.

**Table 1 ijms-24-13987-t001:** Molecular weight (M_W_) distribution of filtrate samples of whey/colostrum protein hydrolysates (WH–5 kDa/CH–5 kDa).

Sample Name	Particle Size (Radius, R), nm	M_W_ of Particles (M_W_ R), kDa	Range of Detected M_W_, kDa	Relative Polydispersity (Pd) ^2^, %	Peptide Length, AA
WH–5 kDa	0.7 ^1^	1.7 ^1^	1.4–2.3 ^3^	48.6	15 ^1^/13–21 ^3^
CH–5 kDa	0.6 ^1^	0.9 ^1^	0.7–1.4 ^3^	41.8	8 ^1^/6–13 ^3^

^1^—parameter for predominant particles, ^2^—parameter of the heterogeneity degree of the system by M_W_ (the ratio of the standard deviation to the mean value, %), ^3^—predominant peptide fraction. The average M_W_ of amino acids (AA) is 110 Da.

**Table 2 ijms-24-13987-t002:** Thermal oxidative destruction parameters of ultrafiltered hydrolysate samples and corresponding complexes based on DTG/DSC profiles (in the area of the dominant peak appropriate for β/γ-CD thermodegradation).

Sample Name	Temperature of Maximum Thermal Oxidative Degradation Speed (T_Vmax_), °C	Maximum Thermal Oxidative Degradation Speed (V_max_), mg/°C	Activation Energy (E_a_), kJ/mol
β/γ-cyclodextrin	301.8 ± 2.0 ^A1^/303.4 ± 2.3 ^A2^	0.43 ± 0.02 ^A1,α^/0.33 ± 0.01 ^A2,β^	324 ± 10 ^A1,α^/292 ± 15 ^A2,β^
Ultrafiltered whey hydrolysate (WH–5 kDa)	266.0 ± 2.0	0.031 ± 0.002	79 ± 1
Mechanical mixture (WH–5 kDa:β/γ-CD = 1:2)	297.5 ± 2.1 ^B1^/289.9 ± 2.5 ^B2^	0.29 ± 0.01 ^B1^/0.17 ± 0.01 ^B2^	118 ± 3 ^B1^/110 ± 1 ^B2^
Complex (WH–5 kDa:β/γ-CD = 1:2)	305.1 ± 2.2 ^C1^/296.6 ± 2.3 ^C2^	0.15 ± 0.02 ^C1^/0.11 ± 0.01 ^C2^	105 ± 3 ^C1,α^/97 ± 2 ^C2,β^
Ultrafiltered colostrum hydrolysate (CH–5 kDa)	285.8 ± 2.9	0.027 ± 0.001	65 ± 1
Mechanical mixture (CH–5 kDa:β/γ-CD = 1:2)	289.7 ± 2.7 ^B1^/283.5 ± 2.3 ^B2^	0.30 ± 0.02 ^B1^/0.18 ± 0.01 ^B2^	125 ± 4 ^B1^/114 ± 3 ^B2^
Complex (CH–5 kDa:β/γ-CD = 1:2)	289.7 ± 3.5 ^B1^/294.3 ± 2.8 ^C2^	0.14 ± 0.01 ^C1^/0.15 ± 0.01 ^C2^	107 ± 2 ^C1,α^/102 ± 1 ^C2,β^

The data for samples with β-CD were partially presented in the article by Halavach et al. (2022) [27]. The values represent the mean ± the half-width of 95% confidence interval (n = 3). Means without a common symbol within the same column (A1–C1, A2–C2) and row (α, β) indicate significant difference at *p* < 0.05.

**Table 3 ijms-24-13987-t003:** Molecular weight distribution of the mixture containing ultrafiltered whey protein hydrolysate (WH–5 kDa) and β/γ-CD depending on incubation time.

Sample Name	Particle Size (Radius, R), nm	Molecular Weight of Particles (M_W_ R), kDa	Relative Polydispersity (Pd) ^2^, %
WH–5 kDa	0.7 ^1^	1.7 ^1^/1.4–2.3	48.6
β-CD	0.7	1.5	81.8
γ-CD	0.8	2.1	37.4
WH–5 kDa:β-CD = 1:0.5, incubation time			
15 min	0.7–1.0	1.5–3.1	30.7–58.0
30 min	0.6–1.2	1.2–5.2	16.9–42.8
60 min	0.6–1.3	1.0–6.7	24.6–42.5
WH–5 kDa:β-CD = 1:1, incubation time			
15 min	0.7–1.0	1.5–3.6	34.3–51.2
30 min	0.7–1.4	1.6–7.2	17.5–45.4
60 min	0.7–1.1	1.7–4.5	31.4–53.0
WH–5 kDa:β-CD = 1:2, incubation time			
15 min	0.7–1.1	1.2–4.5	15.6–42.8
30 min	0.7–0.8	1.6–1.8	32.8–39.1
60 min	0.6–0.7	0.9–1.5	35.2–45.6
WH–5 kDa:γ-CD = 1:0.5, incubation time			
15 min	0.8–1.1	2.1–4.6	27.7–33.5
30 min	0.6–1.0	1.1–3.5	19.5–28.7
60 min	0.8	1.9–2.1	32.7–34.4
WH–5 kDa:γ-CD = 1:1, incubation time			
15 min	0.8–1.1	2.0–3.9	19.3–36.2
30 min	0.8	1.9–2.2	16.0–46.7
60 min	0.6–1.0	1.1–3.2	32.0–59.7
WH–5 kDa:γ-CD = 1:2, incubation time			
15 min	0.6–0.8	1.1–1.9	32.8–45.5
30 min	0.7–0.8	1.5–2.1	26.7–34.7
60 min	0.7	1.3–1.7	32.9–38.7

^1^—parameter for predominant particles; ^2^—parameter of the heterogeneity degree of the system by M_W_ (the ratio of the standard deviation to the mean value, %).

**Table 4 ijms-24-13987-t004:** Parameters of antioxidant activity of dairy protein hydrolysates and related filtrates, β/γ CDs and their complexes with peptides.

Sample Name	IC_50_, μg of Protein/mL	IC_50_ (Native Protein)/ IC_50_ (Hydrolysate)	IC_50_ (Hydrolysate)/ IC_50_ (Complex)
WP	81.6 ± 2.1 ^A^	1.0 ^A^	–
WH	32.1 ± 1.6 ^B^	2.54 ± 0.18 ^B^	–
WH–5 kDa	13.8 ± 0.7 ^C^	5.91 ± 0.16 ^C^	1.0 ^A^
β-CD + WH–5 kDa	6.36 ± 0.35 ^D^	12.9 ± 1.0 ^D,α^	2.18 ± 0.23 ^B,α^
γ-CD + WH–5 kDa	4.97 ± 0.17 ^E^	16.4 ± 0.2 ^E,I^	2.78 ± 0.06 ^C,I^
CP	56.9 ± 0.9 ^a^	1.0 ^a^	–
CH	17.3 ± 0.4 ^b^	3.29 ± 0.05 ^b^	–
CH–5 kDa	8.60 ± 0.24 ^c^	6.62 ± 0.22 ^c^	1.0 ^a^
β-CD + CH–5 kDa	5.95 ± 0.38 ^d^	9.60 ± 0.69 ^d,β^	1.45 ± 0.09 ^b,β^
γ-CD + CH–5 kDa	4.01 ± 0.08 ^e^	14.2 ± 0.4 ^e,II^	2.14 ± 0.13 ^c,II^
β-CD	106.5 ± 0.7 ^F/f^	–	–
γ-CD	76.4 ± 4.0 ^G/g^	–	–

WP—whey proteins, WH—hydrolysate of whey proteins, CP—colostrum proteins, CH—hydrolysate of colostrum proteins, 5 kDa—ultrafiltration with a permeability of 5 kDa. The values represent the mean ± the half-width of 95% confidence interval (n = 3). Means without a common letter (A–G, a–g; α, β; I, II) within the same column indicate significant difference at *p* < 0.05.

**Table 5 ijms-24-13987-t005:** Antigenic properties of native and hydrolyzed whey and colostrum protein samples, β/γ-CD complexes with peptide fractions.

Sample Name	Residual Antigenicity (RA), %	RA (Hydrolysate)/ RA (Complex)
WP	100 ^A^	–
WH	5.8 ± 0.2 ^B^	–
WH–5 kDa	0.047 ± 0.002 ^C^	1.0 ^A^
β-CD + WH–5 kDa	0.0083 ± 0.0003 ^D^	5.62 ± 0.01 ^B^
γ-CD + WH–5 kDa	0.0089 ± 0.0003 ^D^	5.28 ± 0.08 ^B^
CP	100 ^a^	–
CH	0.68 ± 0.05 ^b^	–
CH–5 kDa	0	–
β-CD + CH–5 kDa	0	–
γ-CD + CH–5 kDa	0	–
β/γ-ЦД	0	–

WP—whey proteins, WH—hydrolysate of whey proteins, CP—colostrum proteins, CH—hydrolysate of colostrum proteins, 5 kDa—ultrafiltration with a permeability of 5 kDa. The values represent the mean ± the half-width of 95% confidence interval (n = 3). Means without a common letter (A–D; a, b) within the same column indicate significant difference at *p* < 0.05.

**Table 6 ijms-24-13987-t006:** Antimutagenic effect of native and hydrolyzed whey and colostrum samples, β/γ-CD complexes with peptide fractions in Ames test.

Sample Name	Level of Mutation Reducing (%) (0.03–0.5 mg of Protein Per Plate)	
	*S. typhimurium* TA 98	*S. typhimurium* TA 100
WH–5 kDa	14.10–25.93 ^A,α,I^	12.05–20.19 ^A,α,II^
β-CD + WH–5 kDa	14.07–28.37 ^A,I^	12.34–21.61 ^A,I^
γ-CD + WH–5 kDa	16.98–29.86 ^A,I^	13.92–22.97 ^A,II^
CH–5 kDa	9.15–23.91 ^a,α,I^	8.48–19.36 ^a,α,I^
β-CD + CH–5 kDa	10.00–25.25 ^a,I^	13.69–21.21 ^a,I^
γ-CD + CH–5 kDa	7.91–21.28 ^a,I^	9.95–19.47 ^a,I^
β/γ-ЦД	0	0

WP—whey proteins, WH—hydrolysate of whey proteins, CP—colostrum proteins, CH—hydrolysate of colostrum proteins, 5 kDa—ultrafiltration with a permeability of 5 kDa. The values represent the means (n = 3). Means without a common letter within the same column (A, a, α) and row (I, II) indicate significant difference at *p* < 0.05.

## Data Availability

Data are contained within the article.

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
