# Peer review of "Influence of Complexation with β- and γ-Cyclodextrin on Bioactivity of Whey and Colostrum Peptides"

_ijms, 2023, doi:10.3390/ijms241813987_

Round 1
Reviewer 1 Report
This manuscript exploits the hydrolyzates of whey and colostrum, followed by complexation with cyclodextrins for a bioactivity evaluation. However manuscript contains needs extensive major revisions for publishing.
Introduction:
1) Page 2, lines 77 and 80. Please change “Try” with “Trp” since the authors refer to tryptophan aminoacid.
2) Page 2, lines 88-83. The authors state that Phe, Trp, and Tyr interact with the ß-CD cavity mainly by hydrophobicity. Please also report the case of the other aminoacids involved in the bitterness.
Results:
1) On Pages 3 and 4 the authors report the HPLC chromatograms of native, hydrolyzed and filtrated proteins and state that: “The fractionation leads to the removal of residual amounts of non-cleaved proteins and polypeptides with antigenic potential from enzymatic hydrolysates”. The cut-off of 5kDa should remove samples with Mw lower than 5kDa, which contrasts with the MALDI spectra, where the peaks at 12 kDa are absent. These sections are confusing, please clarify.
2) Page 7, Table 2: The authors report the thermal degradation parameters of samples. Are they referring to the unfiltrate or to filtrate samples? There is a discrepancy in the table.
3) Page 8, Figure 5: The authors report WH in the figure, but WH-5KDa in the table. Please be clear.
Moreover, please separate the figure since it should be clearer to show the complexes with ß- and γ-CDs separately.
4) Page 8, Figure 5: The authors report CH in the figure, but CH-5KDa in the table. Please be clear.
Moreover, please separate the figure since it should be clearer to show the complexes with ß- and γ-CDs separately.
Discussion:
Page 16, lines 425-451 and page 17, lines 452-464: These sections are entirely isolated from the data reported.
Author Response
Response to Reviewer 1 Comments
Authors would like to express sincere gratitude to reviewer for the scrupulous expertise of the paper and valuable comments.
Point 1:
This manuscript exploits the hydrolyzates of whey and colostrum, followed by complexation with cyclodextrins for a bioactivity evaluation. However manuscript contains needs extensive major revisions for publishing.
Introduction:
Point 1.1: Page 2, lines 77 and 80. Please change “Try” with “Trp” since the authors refer to tryptophan aminoacid.
Response 1.1: This comment has been corrected in the text of the article (l. 77 and 80). (in blue)
Point 1.2: Page 2, lines 88-83. The authors state that Phe, Trp, and Tyr interact with the ß-CD cavity mainly by hydrophobicity. Please also report the case of the other aminoacids involved in the bitterness.
Response 1.2: This comment has been corrected in the text of the article (l. 83–87). (in blue)
Point 2:
Results:
Point 2.1: On Pages 3 and 4 the authors report the HPLC chromatograms of native, hydrolyzed and filtrated proteins and state that: “The fractionation leads to the removal of residual amounts of non-cleaved proteins and polypeptides with antigenic potential from enzymatic hydrolysates”. The cut-off of 5kDa should remove samples with Mw lower than 5kDa, which contrasts with the MALDI spectra, where the peaks at 12 kDa are absent. These sections are confusing, please clarify.
Response 2.1: Figures 3 and 4 overlap the spectra of hydrolysates (Figure 3A, 4A) and ultrafiltrates (Figure 3B, 4B). Peaks with MW more than 10 kDa were detected on the mass spectra of the hydrolysates (Figure 3A, 4A). Corrected references to the Figures 3 and 4 are presented in the text of the article (l. 128, l. 134 and 135).
Point 2.2: Page 7, Table 2: The authors report the thermal degradation parameters of samples. Are they referring to the unfiltrate or to filtrate samples? There is a discrepancy in the table.
Response 2.2: This comment has been corrected in the text of the article (l. 184, Table 2). (in blue)
Point 2.3: Page 8, Figure 5: The authors report WH in the figure, but WH-5KDa in the table. Please be clear.
Moreover, please separate the figure since it should be clearer to show the complexes with ß- and γ-CDs separately.
Response 2.3: This comment has been corrected in Figure 5. This figure is divided into 2 parts.
Point 2.4: Page 8, Figure 5: The authors report CH in the figure, but CH-5KDa in the table. Please be clear.
Moreover, please separate the figure since it should be clearer to show the complexes with ß- and γ-CDs separately.
Response 2.4: This comment has been corrected in Figure 6. This figure is divided into 2 parts.
Point 3:
Discussion: Page 16, lines 425-451 and page 17, lines 452-464: These sections are entirely isolated from the data reported.
Response 3: The referred fragment is included in the Discussion section to review the available data on the parameters for obtaining CD inclusion complexes with protein hydrolysates from various sources. Therefore, we would like to propose that this fragment be left in the manuscript (l. 431–455, l. 456–471).
Reviewer 2 Report
The manuscript on the influence of complexation with cyclodextrins on bioactivity of whey and colostrum peptides is interesting and contains several information of practical importance. There are some minor points to clarify.
I would delete „included” from the title. The studied whey and colostrum peptides are mixtures of a large number of compounds and we cannot be sure that all of them are included.
What was the solvent at the preparation of CD/peptide complexes? Were the hydrolysates as obtained in PBS used?
Explain system polydispersity! Polydispersity index is the generally accepted term for characterizing the polydispersity. In the text (line 144) high polydispersity index of the system is mentioned, but usually polydispersity index is between 1.0-2.0.
Reconsider the sentence in lines 166-168: the thermal degradation of B-CD starts at lower temperature and at higher velocity, than G-CD. Why do you think B-CD has a greater thermal stability?
Too much curves are demonstrated in Fig. 5 and 6. Consider to depict the curves belonging to the two CDs to include in two different figures (Fig. 5a and b, Fig. 6a and b). Now it is difficult to compare the curves.
Is B-CD really sweeter than G-CD? Is it the experience of the authors? If not, give reference. I would think the opposite, as the hydrolysis of G-CD starts already in the mouth and the forming lower saccharides might easier reach the taste percepting buds in the oral cavity.
Table 3: Chek the beta- and gamma-signals. Give a few characteristic molecular weight distribution curves.
In the Materials, please, specify if the dairy products are of bovine origin.
Minor remarks:
Line 63: develop new food products (delete of)
Line 99-100: biological activity (not biologically active properties)
Line 123: more than 10 kDa
Table 2, first row: beta is missing, in table note: data (not date)
Line 210: its better solubility (not their)
Line 217: determining the optimal conditions (no need of the second of)
Line 237: a small molar excess
Table 5: check the last line (font type)
Line 463: a several differences (delete a)
Line 619: delete one of the commas
Line 795. method
Line 800: potential 2x in a row (rephrase)
Author Response
Response to Reviewer 2 Comments
Authors would like to express sincere gratitude to reviewer for the scrupulous expertise of the paper and valuable comments.
Point 1:
The manuscript on the influence of complexation with cyclodextrins on bioactivity of whey and colostrum peptides is interesting and contains several information of practical importance. There are some minor points to clarify.
I would delete „included” from the title. The studied whey and colostrum peptides are mixtures of a large number of compounds and we cannot be sure that all of them are included.
Response 1: This comment has been corrected in the text of the article (l. 2 and 3). (in green)
Point 2:
What was the solvent at the preparation of CD/peptide complexes? Were the hydrolysates as obtained in PBS used?
Response 2: This comment has been corrected in the text of the article (l. 638, l. 651 and 652, l. 663, l. 670, l. 721, l. 723, l. 725, l. 731, l. 735). (in green)
Point 3:
Explain system polydispersity! Polydispersity index is the generally accepted term for characterizing the polydispersity. In the text (line 144) high polydispersity index of the system is mentioned, but usually polydispersity index is between 1.0-2.0.
Response 3: Relative polydispersity (%Pd) parameter has been used to evaluate the heterogeneity of the system by molecular weight (MW). %Pd is calculated as the coefficient of variation, or the ratio of the standard deviation to the mean value.
This comment has been corrected in the text of the article (Table 1, l. 146, l. 149 and 150, Table 3). (in green)
Point 4:
Reconsider the sentence in lines 166-168: the thermal degradation of B-CD starts at lower temperature and at higher velocity, than G-CD. Why do you think B-CD has a greater thermal stability?
Response 4: This sentence contains an incorrect conclusion, so it reads as follows: “The β/γ‑CD sample is characterized by a mass loss peak at 301.8/303.4 °C with a maximum thermal destruction rate reaching 0.43/0.33 mg/°C.”(l. 171 and 172) (in green)
“The activation energy of β‑CD was 324 kJ/mol, whereas that of γ‑CD was 292 kJ/mol, that confirms the greater thermal stability (increase in the activation energy of thermal oxidative degradation) of the β‑form”. (l. 203–205) This suggestion provides a conclusion on the thermostability of CDs based on the determination of activation energy.
Point 5:
Too much curves are demonstrated in Fig. 5 and 6. Consider to depict the curves belonging to the two CDs to include in two different figures (Fig. 5a and b, Fig. 6a and b). Now it is difficult to compare the curves.
Response 5: This comment has been corrected in Figures 5 and 6. This figures are divided into 2 parts.
Point 6:
Is B-CD really sweeter than G-CD? Is it the experience of the authors? If not, give reference. I would think the opposite, as the hydrolysis of G-CD starts already in the mouth and the forming lower saccharides might easier reach the taste percepting buds in the oral cavity.
Response 6: The conclusion on the degree of β/γ‑CD cyclodextrin sweetness is based on oun experiments, the corresponding article is in press.
The related sentence reads as follows: “The advantages of β‑CD are its relative thermostability, a simple chemical synthesis scheme, and an available price category.” (l. 218–220).
Point 7:
Table 3: Chek the beta- and gamma-signals. Give a few characteristic molecular weight distribution curves.
Response 7: Thanks to a valuable comment, additional corrections have been included in Table 3. (in green).
Due to the polycomponent composition of WH–5 kDa and close values of particle size (0.7–0.8 nm) for WH–5 kDa и CDs samples, graphical demonstration of DLS results is overloaded and difficult to perceive. Some characteristic DLS profiles of β/γ‑cyclodextrin, mixtures (WH–5 kDa + CDs) are presented below (Figures 1–3).
The DLS results are most clearly and comprehensibly presented in the form of Table 3, which allows us to compare samples of hydrolysate WH–5 kDa and its complexes with CDs, for which a relatively high degree of polydispersity (%Pd) is shown. In this regard, we ask to approve the presentation of the DLS-analysis results in the table format.
Point 8:
In the Materials, please, specify if the dairy products are of bovine origin.
Response 8: This comment has been corrected in the text of the article (l. 584 and 585, l. 603). (in green)
Point 9:
Minor remarks:
Line 63: develop new food products (delete of)
Line 99-100: biological activity (not biologically active properties)
Line 123: more than 10 kDa
Table 2, first row: beta is missing, in table note: data (not date)
Line 210: its better solubility (not their)
Line 217: determining the optimal conditions (no need of the second of)
Line 237: a small molar excess
Table 5: check the last line (font type)
Line 463: a several differences (delete a)
Line 619: delete one of the commas
Line 795. method
Line 800: potential 2x in a row (rephrase)
Response 9: This comment has been corrected in the text of the article (l. 63, l. 103, l. 127, Table 2, l. 216, l. 223, l. 243, Table 5, l. 469 and 470, l. 626, l. 804, l. 808 according to subparagraphs of Point 9). (in green)
Reviewer 3 Report
The manuscript submitted for review by T.M. Halavach et al. describes a relatively interesting, applied problem. I have two main objections to the manuscript in terms of nomenclature and data presentation. I did not encounter the term cyclodextrin complexes as nanocomplexes in the literature. This term appears in the title of the grant through which this research was conducted. It is a pity that the reviewers did not catch this incorrect term earlier. In addition, inclusion complexes of cyclodextrins are not called clathrates. Rather, the word clathrate refers to the crystallographic structure of the crystals. And I also have not encountered the term clathrate applied to cyclodextrin complexes anywhere in the literature. Moreover, cyclodextrins are not the same as cyclic oligosaccharides. For example, there can be cyclic oligosaccharides made of other monosaccharides than glucose, and then they are not cyclodextrins. Alpha- and gamma-CDs also have rigid molecules like beta-CD. This sentence, therefore, is incorrect: At the same time, β-CD is a rigid structure due to the presence of stabilizing hydrogen bonds, thus accounts for its relatively low solubility. (l. 49 and 50). In Table 2, the data and their uncertainties are correctly presented. That is, the uncertainty and the result have the same decimal expansion and the uncertainty has a maximum of two significant digits. By contrast, in Tables 4 and 6, although the decimal expansion is the same, the uncertainties have three and four significant digits, respectively. This should be corrected. The commonly used abbreviation for acetonitrile is ACN rather than AcN (l. 594).
I found no glaring spelling or grammatical errors, but the style needs to be corrected by a professional proofreader, preferably a native speaker.
Author Response
Response to Reviewer 3 Comments
Authors would like to express sincere gratitude to reviewer for the scrupulous expertise of the paper and valuable comments.
Point 1: The manuscript submitted for review by T.M. Halavach et al. describes a relatively interesting, applied problem. I have two main objections to the manuscript in terms of nomenclature and data presentation.
1.1 I did not encounter the term cyclodextrin complexes as nanocomplexes in the literature. This term appears in the title of the grant through which this research was conducted. It is a pity that the reviewers did not catch this incorrect term earlier. In addition, inclusion complexes of cyclodextrins are not called clathrates. Rather, the word clathrate refers to the crystallographic structure of the crystals. And I also have not encountered the term clathrate applied to cyclodextrin complexes anywhere in the literature.
1.2 Moreover, cyclodextrins are not the same as cyclic oligosaccharides. For example, there can be cyclic oligosaccharides made of other monosaccharides than glucose, and then they are not cyclodextrins.
1.3 Alpha- and gamma-CDs also have rigid molecules like beta-CD. This sentence, therefore, is incorrect: At the same time, β-CD is a rigid structure due to the presence of stabilizing hydrogen bonds, thus accounts for its relatively low solubility. (l. 49 and 50).
Response 1:
1.1 The terms "nanocomplex" and "clathrate" have been changed to "complex" and "nanoparticle" throughout the article. (in red)
1.2 The terms "cyclodextrin" and "cyclic oligosaccharide" are used as synonyms on the basis that cyclodextrins, consisting of a macrocyclic ring of glucose residues linked by α‑1,4 glycosidic bonds, are a family of cyclic oligosaccharides.
1.3 This comment has been corrected in the text of the article (l. 48–51). (in red)
Point 2: In Table 2, the data and their uncertainties are correctly presented. That is, the uncertainty and the result have the same decimal expansion and the uncertainty has a maximum of two significant digits. By contrast, in Tables 4 and 6, although the decimal expansion is the same, the uncertainties have three and four significant digits, respectively. This should be corrected.
Response 2: This comment has been corrected in the text of the article (Tables 4–6). (in red)
Point 3: The commonly used abbreviation for acetonitrile is ACN rather than AcN (l. 594).
Response 3: This comment has been corrected in the text of the article (l. 601 and 609). (in red)
The corrections to improve English style and further identify deficiencies are highlighted in orange. (l. 24, 27, 30, 43–45, 53, 57, 58, 178, 179, 192, 193, 251, 252, 271, 272, 278, 279, 301, 302, 335, 336, 411, 503, 536, 538, 558, 720–723, 731, 735, 747, 763, 764, 767–769, 773, 775, 798–800, 904)
Round 2
Reviewer 1 Report
The authors have modified the manuscript accordingly.
Author Response
Authors would like to express sincere gratitude to Reviewer for valuable comments and positive review.
With great respect and appreciation,
Tatsiana M. Halavach et al.
Reviewer 3 Report
I disagree that the terms cyclodextrins and cyclic oligosaccharides are synonymous. Cyclodextrins are cyclic oligosaccharides made exclusively of glucose and linked by an alpha-1,4-glycosidic bond. Thus, the term cyclodextrin has a narrower meaning than cyclic oligosaccharides. Therefore, I believe that the sentence (l. 35 and 36): Cyclodextrins (CDs), or cyclic oligosaccharides, belong to a unique class of molecules that are naturally formed as a result of starch cleavage., is incorrect. I also disagree with the statement that the b-cyclodextrin molecule is made inflexible by the formation of intramolecular hydrogen bonds. The sentence (l. 48-51): At the same time, β-CD has relatively low solubility due to the presence of stabilizing intramolecular hydrogen bond, which leads to inflexibility of the CD structure and reduced ability to form intermolecular hydrogen bond with ambient water molecules., should be corrected. In the molecules of alpha- and gamma-cyclodextrins, intramolecular hydrogen bonds are not formed and their molecules are nevertheless rigid. Sentence (l. 57 and 58): Natural oligosaccharides (α-, β-, γ-CD) do not have a toxic effect on the human organism when administered orally., would indicate that this refers to all possible natural oligosaccharides, and native cyclodextrins are given only as examples. Please clarify. The terms inclusion complexes (or complexes) and nanoparticles are not synonymous. However, I see that these two terms are used interchangeably in the manuscript. In cyclodextrin chemistry, the term nanoparticle is not used to refer to inclusion complexes. In Figures 5B and 6B for gamma-cyclodextrin, the numerical values have commas instead of periods.
In my opinion, the proofreading performed as standard by the publisher is sufficient to ensure the linguistic correctness of the manuscript.
Author Response
Response to Reviewer 3 Comments (Round 2)
Authors would like to express sincere gratitude to reviewer for the scrupulous expertise of the paper and valuable comments.
Point 1: I disagree that the terms cyclodextrins and cyclic oligosaccharides are synonymous. Cyclodextrins are cyclic oligosaccharides made exclusively of glucose and linked by an alpha-1,4-glycosidic bond. Thus, the term cyclodextrin has a narrower meaning than cyclic oligosaccharides. Therefore, I believe that the sentence (l. 35 and 36): Cyclodextrins (CDs), or cyclic oligosaccharides, belong to a unique class of molecules that are naturally formed as a result of starch cleavage., is incorrect.
Response 1: This sentence contains an incorrect phrase, so it reads as follows: “Cyclodextrins (CDs) belong to a unique class of molecules that are naturally formed as a result of starch cleavage.”(l. 35 and 36) (in red)
The term "cyclic oligosaccharide(s)" has been changed to "cyclodextrin(s)" by the text of the article. (l. 57, 97, 184, 199, 217, 275, 283, 296, 320, 388, 428, 641, 733, 754, 774, 797) (in red)
Point 2: I also disagree with the statement that the b-cyclodextrin molecule is made inflexible by the formation of intramolecular hydrogen bonds. The sentence (l. 48-51): At the same time, β-CD has relatively low solubility due to the presence of stabilizing intramolecular hydrogen bond, which leads to inflexibility of the CD structure and reduced ability to form intermolecular hydrogen bond with ambient water molecules., should be corrected. In the molecules of alpha- and gamma-cyclodextrins, intramolecular hydrogen bonds are not formed and their molecules are nevertheless rigid.
Response 2: This sentence includes an incorrect conclusion, so it reads as follows: “Moreover, the water solubility of α‑, β‑, and γ‑CDs is different.”(l. 48 and 49) (in red)
Point 3: Sentence (l. 57 and 58): Natural oligosaccharides (α-, β-, γ-CD) do not have a toxic effect on the human organism when administered orally., would indicate that this refers to all possible natural oligosaccharides, and native cyclodextrins are given only as examples. Please clarify.
Response 3: There are inaccuracies in these sentences, so they read as follows: “α‑, β‑, and γ‑CD do not have a toxic effect on the human organism when administered orally. There is a general trend at the legislative level toward wider recognition of these CDs as food additives [8–10].” (l. 55–57) (in red)
Point 4: The terms inclusion complexes (or complexes) and nanoparticles are not synonymous. However, I see that these two terms are used interchangeably in the manuscript. In cyclodextrin chemistry, the term nanoparticle is not used to refer to inclusion complexes.
Response 4: The term "nanoparticle(s)" has been changed to "complex(es)" (or other phrases) by the text of the article. (l. 20, 21, 28, 29, 89, 96, 99–101, 157, 180, 190, 202, 215, 255, 272, 277, 283, 294, 316, 329, 456, 461, 467, 501, 513, 536, 557, 594, 689, 799, 800, 803, 804) (in red)
Point 5: In Figures 5B and 6B for gamma-cyclodextrin, the numerical values have commas instead of periods.
Response 5: This comment has been corrected in the text of the article (Figures 5B and 6B).